# Identification of HIV-1 Envelope Mutations that Enhance Entry Using Macaque CD4 and CCR5

**DOI:** 10.3390/v12020241

**Published:** 2020-02-21

**Authors:** Jeremy I. Roop, Noah A. Cassidy, Adam S. Dingens, Jesse D. Bloom, Julie Overbaugh

**Affiliations:** 1Divisions of Human Biology, Fred Hutchinson Cancer Research Center, Seattle, WA 98109, USA; jeremyianroop@gmail.com (J.I.R.); ncassidy@fredhutch.org (N.A.C.); adingens@fredhutch.org (A.S.D.); 2Basic Sciences, Fred Hutchinson Cancer Research Center, Seattle, WA 98109, USA; jbloom@fredhutch.org

**Keywords:** HIV, deep mutational scanning, CD4, rhesus macaque

## Abstract

Although Rhesus macaques are an important animal model for HIV-1 vaccine development research, most transmitted HIV-1 strains replicate poorly in macaque cells. A major genetic determinant of this species-specific restriction is a non-synonymous mutation in macaque CD4 that results in reduced HIV-1 Envelope (Env)-mediated viral entry compared to human CD4. Recent research efforts employing either laboratory evolution or structure-guided design strategies have uncovered several mutations in Env’s gp120 subunit that enhance binding of macaque CD4 by transmitted/founder HIV-1 viruses. In order to identify additional Env mutations that promote infection of macaque cells, we utilized deep mutational scanning to screen thousands of Env point mutants for those that enhance HIV-1 entry via macaque receptors. We identified many uncharacterized amino acid mutations in the N-terminal heptad repeat (NHR) and C-terminal heptad repeat (CHR) regions of gp41 that increased entry into cells bearing macaque receptors up to 9-fold. Many of these mutations also modestly increased infection of cells bearing human CD4 and CCR5 (up to 1.5-fold). NHR/CHR mutations identified by deep mutational scanning that enhanced entry also increased sensitivity to neutralizing antibodies targeting the MPER epitope, and to inactivation by cold-incubation, suggesting that they promote sampling of an intermediate trimer conformation between closed and receptor bound states. Identification of this set of mutations can inform future macaque model studies, and also further our understanding of the relationship between Env structure and function.

## 1. Introduction

Entry receptors are a major determinant of the host cell specificity of viruses [1]. Within an infected individual, the distribution of receptors on distinct cell types defines the population of cells susceptible to viral infection. In the case of HIV-1, human T lymphocytes, monocytes and macrophages are the principal host cells due to their expression of the HIV-1 receptor, CD4, and preferred co-receptor, CCR5. Similarly, variation in receptor orthologs among species determines the range of organisms susceptible to viral infection [2]. For HIV-1, although the human CD4 receptor permits efficient infection, CD4 receptors from other nonhuman primates (NHPs) are suboptimal, and only weakly support viral entry [3,4,5,6]. 

The inability of HIV-1 to efficiently utilize CD4 receptors from many NHPs has limited the use of NHPs as models for studying HIV-1 pathogenesis. The Rhesus macaque has been the favored NHP used to date, with SIV/HIV-1 hybrid viruses (SHIVs), which encode HIV-1 Env in place of SIV Env, used to model HIV-1 infection. As SHIVs encoding Envs from circulating HIV-1 strains replicate poorly in macaque cells [7], most studies employing the SHIV/macaque model have used HIV-1 Envs that were lab adapted to enhance macaque infection, and/or that were derived from HIV-1 variants isolated from the chronic stages of infection [7]. These evolved and chronic-stage Envs are highly pathogenic and transmissible in macaques [8]. While these characteristics enabled robust infection of macaques, evolved and chronic-stage Envs display altered structural conformations and increased susceptibility to neutralizing antibodies compared to Envs from circulating HIV-1 strains [8,9]. Use of these Envs in SHIVs therefore compromises the relevance of the SHIV/macaque model for informing HIV-1 infection in humans [7].

In 2012, it was discovered that a single amino acid mutation in macaque CD4 was responsible for the poor ability of HIV-1 to utilize macaque receptors for entry [3]. In vitro evolution experiments also revealed that single amino acid substitutions at sites 204 or 312 in Env’s gp120 subunit allow HIV-1 to more effectively use macaque CD4 as an entry receptor [4]. These mutations disrupted quaternary contacts between Env trimer subunits, and similar to Envs encoded by highly pathogenic SHIVs, increased sensitivity to neutralizing antibodies [10]. 

Identification of these mutants prompted further efforts to identify additional Env mutations that would enhance usage of macaque CD4. In 2016, Li et al. used a structural modeling approach to discover that multiple substitutions at Env site 375 permitted robust infection of macaque cells [11]. Later, in 2018, Del Prete et al. utilized an in vivo evolution strategy to identify a mutation at site 281, located within the CD4 binding site, that enhanced the ability of HIV-1 to use macaque CD4 [12]. Mutations at both 375 and 281 had minimal impact on the trimer structure and antibody neutralization profile. 

The discovery of mutations that allow HIV-1 to use the macaque CD4 receptor without altering neutralization susceptibility has opened the possibility of improving the relevance of the SHIV/macaque model. As these previous studies each utilized a different experimental approach and discovered distinct adaptive mutations, it is likely that additional mutations also exist that would enhance Env usage of macaque CD4. Identification of these mutations could serve to further improve the relevance of the SHIV/macaque model, while also advancing our understanding of the relationship between Env sequence and function. 

Deep mutational scanning (DMS) is a powerful experimental tool that allows researchers to simultaneously assay the effects that thousands of individual mutations have on a protein during selection for a desired function [13]. DMS has previously been used to determine the effects of all single amino acid substitutions in HIV-1 Env in the context of HIV-1 replication in cell culture [14,15], neutralization by antibodies [16,17], and inhibition by fusion inhibitor drugs [18]. Here, we utilized DMS to identify Env mutations that enhance infection of cells expressing macaque CD4 and CCR5 receptors. Importantly, the Env strain background used for these studies was cloned directly from a recently infected individual and is thus representative of transmitted/founder (T/F), circulating HIV-1 strains. We identify a suite of novel mutations in Env’s gp41 subunit that increase HIV-1 infection of cells expressing macaque CD4 and CCR5 receptors.

## 2. Materials and Methods

### 2.1. Cells

HEK293T cells (referred to as 293Ts) were used for production of all replication-competent viruses and pseudoviruses. The 293Ts were maintained in D10 media (Dulbecco’s modified Eagle medium supplemented with 10% heat inactivated fetal calf serum and 2 mM L-glutamine). 

### 2.2. Generation of Stable 293T_rhm_ and 293T_hu_ Cells Lines

Plasmids encoding Rhesus macaque CD4 (rhmCD4) and human CD4 (huCD4) were obtained from [4]. Plasmids encoding Rhesus macaque CCR5 (rhmCCR5) and human CCR5 (huCCR5) were obtained from the NIH-ARP as pBABE.Rh-CCR5 (catalog #3599, GenBank Accession U73739), and pBABE.CCR5 (catalog #3331), respectively. Receptor genes encoded by these four plasmids were each cloned into the pHAGE2 lentiviral vector under a CMV promoter. Transfection of 293Ts with these lentiviral vectors and Fugene-6 (Promega, Madison, WI; E2691) was used to generate virus-like particles (VLPs) carrying receptor gene coding sequences. VLPs carrying rhmCCR5 or huCCR5 genes were used to transduce 293T cells, followed by sorting of cells to isolate populations expressing intermediate levels of either rhmCCR5 (293T_rhmCCR5_) or huCCR5 (293T_huCCR5_). The 293T_rhmCCR5_ cells were then transduced with VLPs carrying the rhmCD4 gene and sorted to isolate 293Ts expressing both rhmCD4 and rhmCCR5 (293T_rhm_). The 293T_hu_ cells were analogously generated by transduction of 293T_huCCR5_ with VLPs carrying the huCD4 gene, followed by sorting. Antibodies used for sorting and quantifying receptor expression were CD4: BD#551980, and CCR5: BD#556042. PCR primers used to amplify and clone receptor sequences are available upon request. 

### 2.3. TZM-bl Titering Assays

Serial dilutions of viral supernatants were added to 2 × 10^4^ TZM-bl cells in the presence of 10 μg/mL DEAE-dextran. After 48 h, cells were fixed and stained for beta-galactosidase, and blue foci were counted to quantify titer. 

### 2.4. Preparation of Green Fluorescent Protein (GFP) Reporter Pseudovirus

The 293T cells were plated at 5 × 10^5^ cells/well in a 6-well dish approximately 24 h prior to transfection. Cells were co-transfected using 6 ul of Fugene-6, 1.2 ug of plasmid Q23ΔEnv-GFP [4], and 8 ug of the Env expression plasmid of interest. Env clones BF520.W14.C2 [19] and BG505.W6.C2.T332N [20] were used for generation of BF520 and BG505 Env pseudoviruses, respectively. Two days after transfection, supernatants were collected and filtered through a 0.2 μM filter, then aliquoted and stored at −80 °C. Supernatants were titered by measuring p24^gag^ levels with an HIV-1 p24 antigen capture assay kit (ABI, Foster City, CA, USA; 5447), and also by TZM-bl assay. 

For generation of Env point mutant pseudoviruses, Env expression plasmids were mutagenized by PCR-mediated site-directed mutagenesis, and Sanger sequencing used to confirm the desired mutation. Primers used for site-directed mutagenesis are available upon request.

### 2.5. Preparation of Replication-Competent BF20.W14M.C2 Viruses

The 293T cells were plated at 5 × 10^5^ cells/well in a 6 well dish approximately 24 h prior to transfection. Cells were co-transfected using 6 μL of Fugene-6, and 2 μg of plasmid Q23.BsmBI.BF520.C2, described in [17]. Two days after transfection, supernatants were collected and filtered through a 0.2 μM filter, then aliquoted and stored at −80 °C. Supernatants were titered by TZM-bl assay. 

### 2.6. Preparation of Mutant Env Libraries

The 293T cells were plated at 5 × 10^5^ cells/well in a 6-well dish approximately 24 h prior to transfection. Cells were transfected using 6 μL of Fugene-6 and 2 μg of BF520.C2 mutant plasmid library, described in [17]. Two days after transfection, supernatants from 16 replicate wells were pooled, filtered through a 2 μM filter, and DNase treated (Roche, Basel, Switzerland; 4716728001) as previously described [14]. Supernatants were titered by TZM-bl assay. 

To establish a genotype–phenotype link for viruses in each replicate mutant library, 1.16 × 10^6^ infectious units of transfection supernatant were used to infect a 3:1 mixture of 293T_hu_ and 293T_rhm_ cells at an MOI of 0.01 in 5 layer flasks (Falcon, Franklin Lakes, NJ, USA; 353144) supplemented with 10 μg/mL DEAE-dextran. Twenty-four hours after infection, the media was replaced with fresh D10 media supplemented with 10 μg/mL DEAE-dextran. Four days after infection, media from all flasks was pooled and filtered through a 0.2 μM filter. Viruses were then concentrated 50-fold by ultracentrifugation over a sucrose cushion at 23,000 RPM for 1 hour at 4°C using a SW28 rotor (Beckman Coulter, Brea California; 342207). Concentrated virus was re-suspended in D10 media, and titered by TZM-bl assay. 

### 2.7. Infection of 293T_rhm_ and 293T_hu_ Cells with Wild-Type BF520 Env

T25 flasks were seeded with 3.3 × 10^6^ of either 293T, 293T_rhm_, 293T_hu_, or a 3:1 mixture of 293T_hu_:293T_rhm_ cells. All flasks were infected with Q23.BsmBI.BF520.C2 virus at an MOI of 0.01. Twenty-four hours after infection, cells were split 1:4. At time points two, three and four days after infection, 60 μL samples were taken for titering on TZM-bl cells.

### 2.8. 293T_rhm_ and 293T_hu_ Selection of Mutant Env Libraries and Deep Sequencing

Wells of a 12-well dish were seeded with 3 × 10^5^ of either 293T_rhm_ or 293T_hu_ cells. The following day, both 293T_rhm_ and 293T_hu_ cells were infected with 1.44 × 10^6^ (replicate 1) or 2.06 × 10^6^ (replicate 2) infectious units of genotype–phenotype linked mutant virus library in the presence of 100 μg/mL DEAE-dextran. We additionally infected one well of 293T_hu_ cells with 1.72 × 10^6^ infectious units of wild-type BF20.W14M.C2 to use as a control during DMS analysis. Three hours after infection, the media was removed and replaced with fresh D10 media. Twelve hours after infection, cells were harvested, pelleted, washed once with PBS, and non-integrated viral DNA was isolated by miniprep. Deep sequencing of viral DNA to quantify the functional effects of Env mutations was performed as described in [17]. 

### 2.9. Env Sequence Numbering

BF520 and BG505 Env residues are numbered according to the reference strain numbering system [21].

### 2.10. Infection of 293T_rhm_ and 293T_hu_ Cells by GFP Reporter Pseudovirus

Wells of a 96-well dish were seeded with 1.2 × 10^4^ of either 293T_rhm_, 293T_hu_, or 293T_rhmCCR5_ cells. The following day, 8 × 10^3^ pg of GFP reporter pseudovirus was added to wells of the plate (corresponding to an MOI of 5) in the presence of 10 μg/mL DEAE-dextran. Two days after infection, cells were harvested, washed once in PBS and re-suspended in 1% paraformaldehyde. Cells were analyzed for GFP expression by flow cytometry. 

### 2.11. Neutralization Assays

Neutralization assays using TZM-bl cells were performed as described in [22]. Reported values reflect the mean of two independent replicates and are representative of at least two neutralization assays. 

### 2.12. Cold Inactivation Assays

Recombinant pseudoviruses were diluted to 1.2 × 10^5^ infectious units/mL, split into 50 μL aliquots, and frozen at −80 °C overnight. At different time points, aliquots were thawed in a 37 °C water bath for two minutes, then placed on ice. Following incubation on ice for different periods of time, all pseudovirus aliquots were incubated in a 37 °C water bath for two minutes, and 6 × 10^2^ IPs were added to wells of a 48-well plate containing 2 × 10^4^ TZM-bl cells in 10 μg/mL DEAE-dextran. Two days later, infection of TZM-bl cells was quantified by a firefly luciferase expression assay [23]. 

### 2.13. Structural Analysis

All structural analyses used the X-ray diffraction model of the BG505 SOSIP trimer in complex with a PGT121 precursor Fab [24]. Molecular graphics and analyses were performed using UCSF Chimera, developed by the Resource for Biocomputing, Visualization, and Informatics at the University of California, San Francisco, with support from NIH P41-GM103311 [25]. 

### 2.14. DMS Data Analysis and Source Code

DMS sequencing data analysis was performed using the dms_tools2 software package version 2.4.1 [26] (https://jbloomlab.github.io/dms_tools2/). A jupyter notebook that describes this analysis and generates Figures 2,3 and 5 is available at: https://github.com/jbloomlab/HIV_Env_macaque_receptor_DMS. The deep sequencing data are available on the Sequence Read Archive under accession numbers SRA: SRX5199658–SRX5199665. 

## 3. Results

### 3.1. Deep Mutational Scanning Identifies Env Mutations That Improve Usage of Macaque Receptors

The Env used in our DMS, subtype A BF520.W14M.C2 *env* (hereafter referred to as BF520), is a (T/F) strain cloned directly from a Kenyan infant at the time point when HIV was first detected following mother-to-child transmission [19]. Like Envs from other circulating HIV-1 strains, the wild type BF520 Env supported infection of cells engineered to express human CD4 and CCR5 (293T_hu_) but was severely restricted in its ability to infect cells expressing macaque receptors (293T_rhm_) (Appendix A) [3]. 

In our DMS, we screened a library of BF520 Env variants to identify mutations that increased entry into cells expressing macaque receptors. The DMS strategy we employed is outlined in Figure 1a, with all steps performed in biological duplicate. Viruses bearing mutant *env* genes were generated by transfection of two independently constructed mutant plasmid libraries into 293T cells. Each plasmid in these replicate plasmid libraries encodes a codon level mutational variant of BF520 *env,* in the context of the proviral genome from the subtype A strain, Q23, which was isolated during the first year of infection [17]. Prior deep sequencing demonstrated that these libraries collectively encode over 95% of the 12,559 possible single amino acid mutations to Env’s ectodomain (residues 32 to 702), with an average of 1.1 codon mutations per each *env* variant [17]. Mutant viruses were then passaged for four days at a low MOI (0.01) in a 3:1 mixture of 293T_hu_ and 293T_rhm_ cells in order to establish a genotype–phenotype link between mutant *env* genes and Env proteins on each virus. A mixture of 293T_hu_ and 293T_rhm_ cells was used for this passage so as to avoid the possibility that *env* mutants that enhanced usage of macaque receptors would drop out of the pool. Importantly, replication kinetics of wild-type BF520 viruses in this mixed cell context were indistinguishable from replication in 293T_hu_ cells alone (Appendix A), indicating that a reduction in susceptible target cells during this step was unlikely to impose a bottleneck on the diversity of the mutant virus pool. Following the low MOI passage, genotype–phenotype linked viruses were used to infect 293T_rhm_ cells in order to select for mutants that enhance entry via macaque receptors. In order to investigate whether mutations enriched by this infection represented macaque-specific adaptations or instead improved infection by more general mechanisms, we simultaneously performed parallel infections of 293T_hu_ cells with the same mutant virus pool. Importantly, expression of CCR5 and CD4 receptors was similar between 293T_rhm_ and 293T_hu_ cell types (Appendix A). 

Following infection of 293T_rhm_ and 293T_hu_ cells, we deep sequenced *env* DNA isolated from infected cells, as well as from the initial mutant plasmid library. Similar to previous DMS experiments, the frequency of stop codons in post-selected samples was reduced to 19–24% of their frequency in the mutant plasmid library (Figure 1b), evidence that our DMS strategy effectively selected for functional Env variants [17,27]. Frequencies of nonsynonymous mutations were also depleted in post-selected samples compared to pre-selected plasmid libraries, consistent with previous reports that amino acid substitutions in Env are generally deleterious (Figure 1b) [14,15]. 

To quantify the effect of each Env mutation on infection of either 293T_rhm_ or 293T_hu_ cells, we first inferred the relative preference for all 20 amino acids at each Env site, for each infection, as previously described [26]. We next calculated a statistic, S_mut/wt_, which describes selection on each Env mutant as the log_2_ of the ratio of the mutant residue preference to the wild-type residue preference at a given site (Figure 1a). S_mut/wt_ values from replicate 293T_hu_ infections were well correlated (Figure 1c) (Pearson *R* = 74), whereas those from replicate 293T_rhm_ infections were more moderately correlated (Figure 1d) (Pearson *R* = 53), although still comparable to previous Env DMS experiments [14]. The weaker correlation between 293T_rhm_ replicates is suggestive of a highly selective bottleneck in which strongly beneficial mutations were enriched against a backdrop of stochastic infection by neutral and slightly deleterious variants (Figure 1d, compare upper right quadrant to other quadrants). As expected, the majority of Env mutations were selected against during both 293T_rhm_ and 293T_hu_ infections, evidence that these were generally detrimental to Env function (Figure 1c–e). 

Infection of 293T_rhm_ and 293T_hu_ cells exerted similar overall selective pressures on Env mutants, as evidenced by the strong correlation between the average S_mut/wt_ values obtained from these infections (Figure 1e). However, the magnitude of enrichment for strongly beneficial mutations was far greater following 293T_rhm_ infection (Figure 1e, points above grey line in upper right quadrant). For example, the most highly enriched mutation following 293T_rhm_ infection (N656Y) was enriched 34-fold relative to the wild-type residue, compared to a 12-fold enrichment for the most strongly selected mutation during 293T_hu_ infection (E482K). This finding likely reflects the fact that wild-type BF520 Env is poorly adapted to infect macaque cells, and therefore mutants in this background have the potential to greatly enhance usage of macaque receptors.

Env mutations that had positive S_mut/wt_ values following infection of 293T_rhm_ cells are shown in Figure 2. Surprisingly, while previous studies have predominately identified mutations in Env’s gp120 subunit that improve infection of macaque cells [4,11,12,28], the majority of mutations highly enriched in our experiments were located in the gp41 subunit. In fact, all amino acid mutations enriched greater than 10-fold over wild-type residues in our DMS were located at just seven sites, all within gp41. Six of these sites were located within either the N terminal heptad repeat (NHR) or C-terminal heptad repeat (CHR), while the seventh was in the intervening loop region. 

While the most strongly enriched mutations were in N/CHR regions, we observed modest enrichment of mutations in several other Env regions. At Env sites 58–63, 20 mutations were selected that improved 293T_rhm_ infection 3- to 8-fold compared to wild-type residues (Figure 2). These sites lie within a short helix that was recently found to make contact with a CD4 molecule bound to the inner domain of a neighboring gp120 protomer [29]. Our DMS also revealed that 293T_rhm_ infection is modestly improved by Env mutations that abolish the putative glycosylation motif at sites 611–613 (Figure 2). 

Given the surprising result that previously uncharacterized mutations in the N/CHR showed the greatest enrichment in the DMS, we sought to validate the effect of these mutations on macaque receptor usage through an independent assay. We generated pseudoviruses bearing a subset of these gp41 mutations in the BF520 Env background, then tested the ability of the pseudovirus mutants to infect 293T_rhm_ cells. Compared to wild-type BF520, all seven gp41 mutants assayed in these experiments improved entry into 293T_rhm_ cells (Figure 3a). Similar to their relative enrichment in the DMS, mutations in the CHR were most beneficial to 293T_rhm_ infection. For example, CHR mutations Q653L and Q652F improved infection of 293T_rhm_ cells 9- and 7-fold, respectively, whereas the most advantageous NHR mutation, I573L, improved infection 4-fold (Figure 3a). These data thus validate that N/CHR mutations enriched in our DMS following infection of 293T_rhm_ cells do indeed enhance entry of BF520 Env into cells bearing macaque receptors.

### 3.2. Similar Mutations Improve Usage of Both Macaque and Human Receptors

Infection of 293T_hu_ cells with the mutant virus pools selected for beneficial mutations at many of the same sites where mutations were enriched following 293T_rhm_ infection (Figure 4). Mutations at sites 58–61, as well as those in the N611 glycosylation motif, were enriched up to 8-fold over wild-type residues following 293T_hu_ infection, similar to their enrichment following 293T_rhm_ infection.

Within gp41, 293T_hu_ infection selected for mutations at many of the same sites as 293T_rhm_ infection, but with a lower magnitude of enrichment. Within the NHR and CHR, no mutations were enriched greater than 10-fold over the wild-type-residue following 293T_hu_ infection, compared to 24 such mutations following 293T_rhm_ infection (Appendix A). This difference in enrichment was especially pronounced at CHR sites 653 and 656, where the most beneficial mutations were enriched 27-and 34-fold, respectively, following 293T_rhm_ infection, but only 5- and 9-fold following 293T_hu_ infection. 

Few mutations outside of the CD4-contacting helix (sites 54–74), the N611 glycan motif, and N/CHR regions were selected for during either 293T_rhm_ or 293T_hu_ infection. However, two notable exceptions were sites 588 and 589, located in the loop region between the NHR and CHR. Following 293T_rhm_ infection, many mutants were enriched up to 21-fold over WT residues at site 588, with a few mutants weakly enriched at site 589. The inverse was true for 293T_hu_ selection, which enriched for mutants up to 6-fold at site 589, but weakly at site 588. 

Pseudovirus infection assays qualitatively validated the effects of mutations enriched in our DMS following 293T_hu_ selection (Figure 3b). Consistent with the DMS data, mutations R557L and K588L, which improved infection of 293T_rhm_ cells, were mildly detrimental in the context of 293T_hu_ infection. Five other N/CHR mutations that were enriched following both 293T_hu_ and 293T_rhm_ DMS selections improved entry of pseudovirus into 293T_hu_ cells in this assay. The magnitude of the benefit to 293T_hu_ infection in this assay was less than in the context of 293T_rhm_ infection (Figure 3a,b), again mirroring the increased effect size of these mutations observed in the DMS. Taken together, these data validate that 293T_rhm_ and 293T_hu_ infection selected for beneficial mutations within similar regions of Env, often at identical sites. 

### 3.3. Mutations Identified by DMS have Smaller Effects than Previously Identified Macaque Adaptive Mutations

Surprisingly, our DMS did not select for Env mutations at sites 204, 281, 312 or 375 that have previously been identified to enhance infection of macaque cells [4,11,12]. Mutations at these sites were adequately represented in the initial mutant plasmid library thus ruling out the possibility that they were not present at the start of our DMS experiment. Several possible hypotheses could explain the apparent lack of selection for these mutants in the DMS. First, these mutations may not have been tolerated in the BF520 Env strain background. If this had been the case, they would have dropped out of the DMS following passage of the mutant plasmid library. A second possibility is that these mutations may not have improved usage of macaque receptors in the BF520 strain background.

To determine whether either of these possibilities explained the lack of enrichment of these previously identified mutants, we first tested whether we could generate infectious pseudovirus bearing two of these mutations, A204E and S375Y, in the BF520 background. The titers of pseudoviruses bearing the S375Y Env mutation, as determined by infecting TZM-bl cells, were similar to those of pseudoviruses bearing wild-type BF520 (2.6 × 10^6^ and 2.8 × 10^6^ infectious units/mL, respectively). In contrast, replicate transfection supernatants containing A204E mutations were not infectious. We thus conclude that the A204E mutation is incompatible with the BF520 Env, thereby explaining its lack of enrichment in our DMS.

We next assessed whether the S375Y mutation in the BF520 background would enhance infection of 293T_rhm_ and 293T_hu_ cells in pseudovirus reporter assays. As shown in Figure 3a, the S375Y mutations increased infection of 293T_rhm_ cells 34-fold compared to wild-type BF520. This was a higher fold increase in infection than any of the mutations that were most enriched in the DMS. In the context of 293T_hu_ infection, S375Y enhanced wild-type infection by 1.7-fold (Figure 3b). These data indicate that the lack of enrichment of S375Y in our DMS cannot be attributed to this mutation being incompatible with the BF520 Env, nor to it impeding 293T_rhm_ infection in this strain background. Additionally, the fact that S375Y was not detrimental to 293T_hu_ infection indicates that the mixture of 293T_hu_ and 293T_rhm_ cells used to establish a mutant genotype–phenotype link in our DMS was likely not responsible for selection against this mutant. 

Why then, was 375Y not enriched in our DMS? An examination of our data revealed that the frequency of S375Y, as well as previously identified macaque adaptive mutations G312V and A281T, was lower following 293T_rhm_ and 293T_hu_ than in the mutant plasmid libraries, evidence of selection against these mutations during the experiment. To further explore this finding, we revisited data from a previously conducted DMS experiment in which the same BF520 mutant Env library was passaged and selected in a human cell line expressing human CD4 and CCR5 (SupT1 T cell line; [17]). In these data, we observed a similar drop in frequencies of S375Y, G312V, and A281T that mirrored the depletion of these mutations in the current DMS experiment. Selection against these mutations in both DMS experiments suggests that properties inherent to the mutant plasmid library or the DMS protocol are likely responsible for their lack of enrichment following 293T_rhm_ infection. One potential explanation is that while these mutations enhance entry of pseudovirus into 293T_rhm_ and 293T_hu_ cells, they are detrimental in the context of multi-cycle replication and purged from the mutant pool during the low MOI infection necessary to generate genotype–phenotype linked virions. 

### 3.4. NHR/CHR Mutants Enhance Usage of Macaque Receptors by Env Strain BG505

We next sought to determine whether the mutations enriched in the DMS were advantageous solely in the context BF520 Env, or instead, reflected a more widely accessible mutational strategy through which Env could evolve to utilize macaque receptors. To investigate this, we introduced seven N/CHR mutants enriched in our DMS into the BG505.W6M.C2.T332N *env* (hereafter referred to as BG505) [30], then tested the ability of these mutants to enhance pseudovirus infection of 293T_rhm_ cells. BG505 is a subtype A, T/F Env [30], 15.8% diverged from BF520 at the amino acid level, and used extensively as a SOSIP trimer for Env structural studies [20,31,32,33,34], and as a vaccine immunogen [20,35]. Recent DMS efforts have also revealed that the site-specific amino acid preferences and mutational tolerance of BG505 Env have diverged from BF520 [15]. 

Infection of 293T_rhm_ cells with wild-type BG505 also revealed this strain to be ~7-fold less effective at entering cells bearing macaque receptors compared to wild-type BF520. At an MOI of 5, only 0.12% of 293T_rhm_ cells were infected following wild-type BG505 infection (Figure 3c) compared to 0.87% following wild-type BF520 infection (Figure 3a). 

Four of the seven N/CHR mutants introduced into BG505 improved the ability of this Env to infect 293T_rhm_ cells (Figure 3c). The magnitude of this effect was smaller than in the BF520 strain, however, with the most advantageous mutant (Q653L) conferring a 2.5-fold improvement over wild-type BG505. Generally, mutations that conferred the greatest improvement to macaque receptor usage in the BF520 strain background also had the largest effect in BG505. For example, mutations Q652F and Q653L, which conferred the greatest benefit to infection in the BF520 background were also the most advantageous in the BG505 strain. Only one mutation, R557L, benefited BF520 infection of 293T_rhm_ cells, but was detrimental to infection of these cells in BG505. Taken together, these data indicate that while some N/CHR mutations have strain-specific effects, the benefit conferred by larger-effect mutations is not restricted to the BF520 background. 

### 3.5. N/CHR Mutations Increase Susceptibility to MPER Antibodies but Minimally Affect Other Epitopes

We next sought to investigate the mechanism through which the N/CHR mutations enhanced usage of macaque receptors. First, we tested whether these mutants conferred a CD4-independent phenotype by assaying whether they allowed entry of BF520 pseudovirus into 293T cells expressing only the macaque CCR5 receptor (293T_rhmCCR5_). No infection was observed by either wild-type or mutants Envs in this assay, thereby ruling out CD4-independence as the mechanism through which N/CHR mutants enhance 293T_rhm_ infection (Figure 3d). 

We next used a panel of monoclonal antibodies (mAbs) and inhibitory mimetic drugs to determine whether N/CHR mutations caused alterations in Env trimer structure that would inform on a mechanism through which they improved entry into 293T_rhm_ cells. Drugs and mAbs in the panel were chosen so as to detect alterations to trimer structure in regions both proximal and distal to the mutant N/CHR residues, with particular interest in the CD4 binding site. In these experiments, we assessed neutralization susceptibility of three mutants that were highly enriched in our DMS: I573L (NHR), Q653L (CHR) and N656Y (CHR). Neutralization of each mutant was assessed in both the BF520 and BG505 Env strains so as to inform on the background dependence of any structural effects. Compared to wild-type Envs, introduction of N/CHR mutants had little effect on neutralization sensitivity to antibodies targeting the CD4bs, trimer apex, V3 loop, or six-helical-bundle epitopes (Figure 5). All tested mutants did, however, increase sensitivity to the MPER antibodies 4E10 and 10E8 in both BG505 and BF520 backgrounds (Figure 5). This effect was strongest for CHR mutants Q653L and N656Y, which increased sensitivity to MPER antibodies by 15- and 8-fold, in BF520 and BG505 backgrounds, respectively (Figure 5a,b). In the BF520 strain, NHR/CHR mutants also modestly increased sensitivity to T-20, a fusion-inhibitor drug that binds a hydrophobic pocket at the N-terminus of the NHR helix (Figure 5a,c). Again, the effect was largest for CHR mutants, which increased T-20 sensitivity 2- to 4-fold.

### 3.6. N/CHR Mutations Increase Sensitivity to Cold Inactivation

Prior to engagement of host receptors, it is thought that Env predominately maintains a closed conformation in which MPER and NHR residues are poorly exposed and not accessible to interactions with mAbs or small molecule inhibitors [36]. Receptor binding initiates formation of the pre-hairpin intermediate conformation, in which both the MPER epitopes, as well as the T-20 binding site in the NHR are exposed [37,38,39]. Several recent publications have described Env mutations that promote unliganded trimer sampling of an intermediate conformation, termed “state 2”, between closed (state 1) and receptor bound (state 3) conformations [40,41,42,43]. State 2 is characterized by a heightened intrinsic reactivity to ligand binding, increased capacity to infect cells expressing low levels of CD4 or CCR5 receptors, and increased exposure of MPER and NHR residues. Given that the N/CHR mutants we identified increase exposure of MPER epitopes, and in strain BF520, the NHR helix, we hypothesized that they may increase reactivity to macaque receptors by promoting sampling of a state 2 conformation. 

Envs frequently sampling state 2 conformations display increased sensitivity to inactivation during cold incubation [40,44]. To investigate whether N/CHR mutants identified in our DMS also increased sensitivity to cold-inactivation, we assessed infectivity of wild-type and mutant BF520 pseudoviruses following incubations on ice. Introduction of Q653L, N656Y and I573L into BF520 decreased Env infectivity following ice incubation (Figure 6). This effect was more pronounced for Q653L and N656Y than for I573L, paralleling the relative effect sizes of these mutants observed during 293T_rhm_ infection and in neutralization sensitivity assays. It thus appears that the degree to which N/CHR mutants enhance usage of macaque receptors may be correlated to increased cold-sensitivity and exposure of MPER epitopes on the unliganded Env trimer. We note that this is not uniformly the case, however, as the N656Y mutation in BG505 increased sensitivity to MPER antibodies but did not enhance infection of 293T_rhm_ cells (Figure 4), evidencing strain-specific effects.

## 4. Discussion

We identified mutations in several regions of an Env from a transmitted HIV-1 virus that led to increased entry using macaque CD4 and CCR5 receptors. Mutations within gp120 modestly improved usage of macaque receptors, but mutations with the largest beneficial effect were located in gp41, most notably in the N/CHR regions. None of the tested mutations conferred a CD4-independent phenotype. Several N/CHR mutations also enhanced entry into cells bearing macaque receptors by a second transmitted Env strain, indicating that their effects were not strain specific. While N/CHR mutations have previously been identified that heighten fusogenic activity [45,46] and increase infectivity of cells expressing low levels of CD4 and CCR5 receptors [40,47], none have been shown to enhance entry of cells bearing macaque receptors. Many of the N/CHR mutations we identified also enhanced entry into cells bearing human CD4 and CCR5 receptors, but the magnitude of this effect was smaller than during infection of cells bearing macaque receptors. 

Our observation that N/CHR mutations increase susceptibility to neutralization by MPER antibodies, and, in the BF520 background, to inhibition by the fusion inhibitor T-20, provide some insight into their effects on Env trimer conformation. The MPER epitope (sites 671–683) and T-20 binding pocket (sites 547–556) are located immediately N, and C terminal, respectively, to the NHR and CHR mutants enriched in our DMS [18,48]. Our neutralization data thus indicate that these mutations cause Env conformation changes that are confined to regions of the trimer directly adjacent to the mutated residues. The mechanism by which these mutants enhance usage of macaque receptors thus appears to be distinct from that of mutations A204E and G312V, which neutralization data suggest disrupt quaternary Env contacts and promote a more open trimer conformation [10]. The mechanism also appears to be distinct from mutations at site 375 and 281, which are located within the CD4 binding site region, and minimally impact Env susceptibility to neutralizing antibodies [11,12]. 

Viruses with mutations in NHR and CHR also demonstrated increased sensitivity to inactivation by cold. This phenotype has been linked to Envs frequently sampling state 2 conformations, in which the thermodynamic barrier between unliganded and CD4-bound states is lowered [40,41]. Envs displaying a state 2 phenotype are also known to display heightened sensitivity to MPER antibodies [40,41,42,43], and in some cases, increased exposure of the NHR helix [40]. Our observations that N/CHR mutations increase sensitivity to MPER antibodies, and in the BF520 background modestly increase T-20 sensitivity, are consistent with the hypothesis that these mutations promote a state 2 conformation. 

Mapping of the N/CHR mutations identified here onto the structure of the unliganded BG505 SOSIP trimer reveals that NHR mutations are located within the central core of the trimer, while CHR mutations are positioned at the trimer base. Mutations in both regions are located at sites that make inter- and intra-protomer contacts with adjacent gp120 and gp41 subunits (Figure 7). Given that previous reports have described N/CHR mutations that modulate trimer reactivity through alterations of inter-protomer contacts [43,49,50], it is possible that the mutations identified here promote a state 2 phenotype by disrupting the stabilizing interactions between protomers within the unliganded trimer. If N/CHR identified here do indeed promote a state 2-like conformation, this would lower the thermodynamic barrier between unliganded and CD4 bound Env states, which could allow relatively weak interactions between Env and macaque CD4 to initiate the conformational changes leading to exposure of the co-receptor binding site. Subsequent binding of the co-receptor would lead to the cascade of Env rearrangements that results in membrane fusion and cell entry. While speculative, this model of macaque receptor usage is in line with a recent report of retroviral host-range expansion associated with envelope mutations that promote a receptor-bound conformation [51]. Alternatively, N/CHR mutations could enhance entry into macaque cells by a distinct mechanism. For example, mutations to Env’s gp41 and heptad repeat regions have previously been found to alter virus infectivity by altering Env expression [52], incorporation into virions [53,54], and trimer stability [55,56]. The N/CHR mutations identified in our DMS may enhance entry into 293T_rhm_ though these or other mechanisms, either independent of, or in conjunction with their potential effects on trimer reactivity.

We note that although the NHR and CHR mutations identified in this work generally enhance usage of both macaque and human receptors, they occur at sites that are highly conserved among circulating HIV-1 strains and lentiviruses [50,57]. Selective pressures exerted by the human immune system could explain the discrepancy between the enrichment of these mutations in our experiments and their lack of representation in natural isolates. Specifically, our data suggest that N/CHR mutations that enhance entry also make the virus more susceptible to neutralization by MPER antibodies. As the MPER epitope is highly conserved [48] and functionally constrained [15], mutations that exposed this region are predicted to be strongly selected against. Thus, in an established HIV-1 infection, the advantage that N/CHR mutations have during entry may be balanced by the necessity of evading the immune response. 

While the most strongly enriched mutations were in N/CHR regions, we observed modest enrichment of mutations in several other Env regions, including sites 58–63 and 611–613. Enriched mutations at sites 58–63 are located within the recently discovered CD4-contacting helix, in which several residues make contact with CD4 bound to the inner domain of an adjacent gp120 protomer [29]. Mutations in this region, when introduced into SOSIP trimers, have also been shown to affect reactivity to CD4 engagement and conformational dynamics [35]. The mutations enriched in our DMS may therefore improve entry by either strengthening contacts with CD4 or increasing Env reactivity to CD4 engagement. The role of mutations at sites 611–613, which abolish the putative N611 glycosylation motif, is less clear. Previous studies have found that removal of certain N-linked glycosylation sites in SIV gp120 enhance replication in macaque macrophages [58] and increase exposure of CD4-contact residues [59], but the relevance of these observations to the loss of the N611 glycan in the context of 293T_rhm_ infection is unclear. Interestingly, the magnitude of mutant enrichment within both the N611 glycan region, and the CD4-contacting helix was similar between 293T_rhm_ and 293T_hu_ selections, suggesting that these mutations function by a mechanism that is similarly advantageous to both human and macaque infection. 

Our DMS did not select for mutations that have previously been found to improve usage of macaque CD4 and CCR5. This observation can be attributed in part to incompatibilities between these mutants and the BF520 Env background, as was observed for A204E. For other mutations, such as S375Y that clearly improved entry of BF520 pseudovirus into 293T_rhm_ cells, the lack of enrichment is harder to explain. One possibility is that these mutations are detrimental in the BF520 background during multi-cycle replication and were removed from the library during the initial low MOI passage. Regardless of the underlying cause, depletion of these mutations indicates that our DMS did not quantify the effects of every possible BF520 Env mutation on the usage of macaque receptors. Future studies that employ different experimental methods are therefore likely to discover additional Env mutants that enhance entry into macaque cells.

A notable limitation of the experiments we performed is that we characterized N/CHR mutants in engineered 293T cells lines that may express CD4 and CCR5 at different levels than human and macaque primary cells. The phenotypes of the Env mutants described here may therefore differ during infection of other cell types, and fitness tradeoffs associated with these mutations may arise during multi-cycle replication in a more clinically relevant context. Additionally, while our data suggest that N/CHR mutations enhance intrinsic Env reactivity, a mechanistic understanding of the effects of these mutations will benefit from further investigations into their effects on CD4 binding and fusion kinetics during infection of relevant cell types. 

This work uncovered many novel determinants of entry into cells bearing both human and macaque receptors. Mutations within the N/CHRs had the largest benefit to usage of receptors from both species and may function by promoting a state 2 Env conformation with heightened reactivity to CD4 binding. Identification of these mutations provides insight into the mechanisms of viral entry and interactions with cellular receptors and may be of use in developing more robust SHIV models.

## Figures and Tables

**Figure 1 viruses-12-00241-f001:**
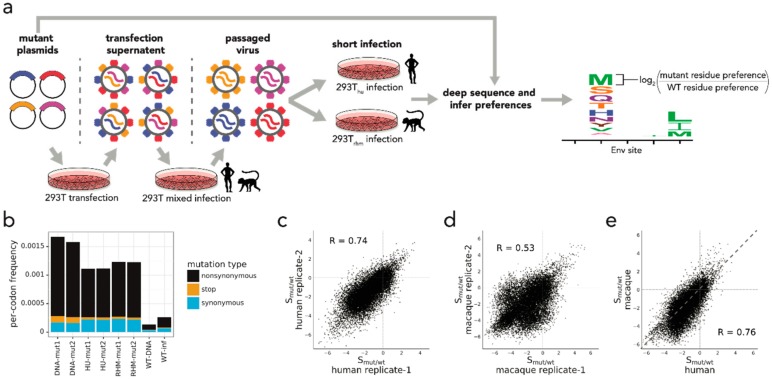
Deep mutational scanning of BF520 Env. (**a**) Schematic of the deep mutational scanning method used. Mutant virus pools were generated by transfection of mutant proviral plasmids followed by low MOI passage. Virus pools were then used to infect either 293T_rhm_ or 293T_hu_ cells. Following infection, amino acid preferences at each Env site were inferred by deep sequencing Env variants from pre- and post-selected samples. Preferences were used to infer S_mut/wt_ values for each mutant as log_2_ (mutant preference / wild-type preference), as depicted by the logoplot. (**b**) The bar graph reports the frequencies of nonsynonymous, stop, and synonymous mutations, as indicated in the key to the right. Labels below each bar indicate the stage of library selection (DNA-mut: mutant plasmid DNA; HU-mut and Rh-mut: mutant virus following selection in 293T_hu_ or 293T_rhm_; WT-DNA: wild-type plasmid DNA control; WT-inf: wild-type virus control following selecting in 293T_hu_ cells). (**c**,**d**) Pearson correlations between S_mut/wt_ values from replicate 293T_hu_ infections (**c**) and 293T_rhm_ infections (**d**). (**e**) Pearson correlation between average S_mut/wt_ values from 293T_rhm_ and 293T_hu_ infections. Points above the diagonal grey line indicate mutants that were more advantageous during 293T_rhm_ infection than during 293T_hu_ infection.

**Figure 2 viruses-12-00241-f002:**
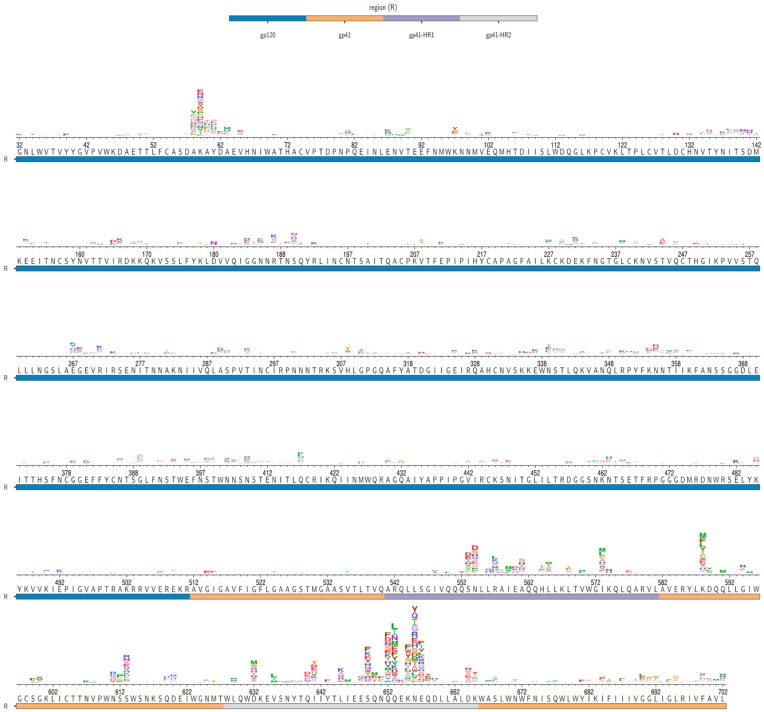
S_mut/wt_ values of amino acids enriched during 293T_rhm_ infection. BF520 Env wild-type amino acid identities are shown below the axes for residues 32–702. Logoplots above the axes indicate mutants that were enriched relative to wild-type residues (S_mut/wt_ > 0) following infection of 293T_rhm_ cells in our DMS. The height of each letter is proportional to the S_mut/wt_ value for that amino acid mutant following 293T_rhm_ infection, averaged over both replicate infections. Site numbers correspond to HXB2 numbering. The underlay bar indicates Env region.

**Figure 3 viruses-12-00241-f003:**
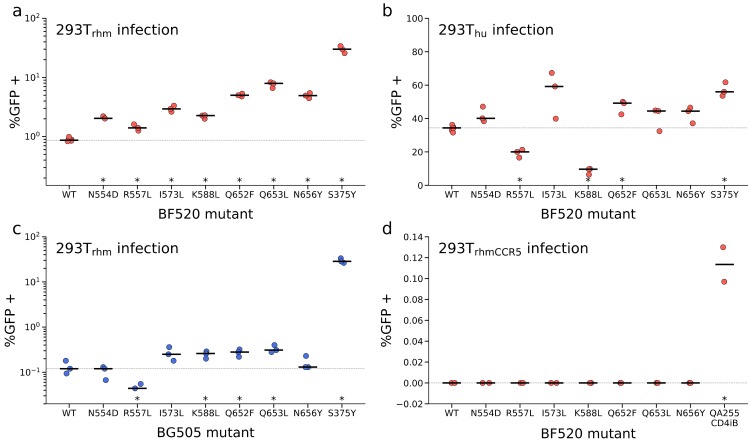
Infection of 293T_rhm_, 293T_CCR5,_ and 293T_hu_ cells by point mutant pseudoviruses. Each panel reports the percentage of the indicated cells that were GFP positive 48 h post-infection (MOI of 5) with the indicated wild-type or mutant viruses. The Env strain in which mutants were engineered is shown at the bottom of each panel. Every point in each panel represents one experimental replicate, and the median, indicated as a horizontal black bar, is representative of at least two independent experiments. Asterisks indicate mutants whose infection was significantly different than the wild-type (*p* < 0.05, Welch’s t-test). (**a**) Infection of 293T_rhm_ by point mutants in the BF520 Env background. (**b**) Infection of 293T_hu_ by point mutants in the BF520 Env background. (**c**) Infection of 293T_rhm_ by point mutants in the BG505 Env background. Note that infection of these cells by wild-type BG505 is nearly 10-fold less efficient than by wild-type BF520. (**d**) Infection of 293T_rhmCCR5_ by point mutants in the BF520 Env background. QA255 CD4iB is a CD4-independent Env described in [10] used here as a positive control.

**Figure 4 viruses-12-00241-f004:**
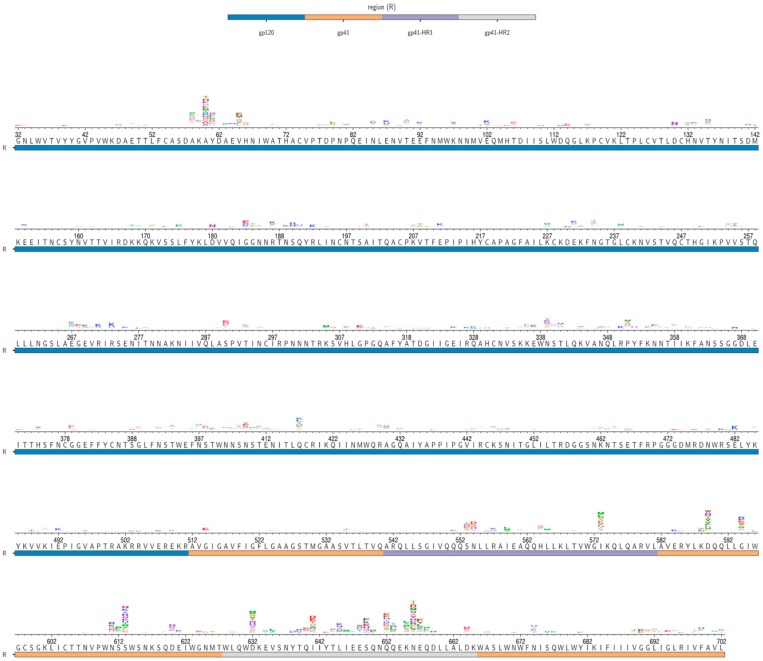
S_mut/wt_ values of amino acids enriched during 293T_hu_ infection. BF520 Env wild-type amino acid identities are shown below the axes for residues 32–702. Logoplots above the axes indicate mutants that were enriched relative to wild-type residues (S_mut/wt_ > 0) following infection of 293T_hu_ cells in our deep mutational scanning (DMS). The height of each letter is proportional to the S_mut/wt_ value for that amino acid mutant following 293T_hu_ infection, averaged over both replicate infections. Site numbers correspond to HXB2 numbering. The underlay bar indicates Env region.

**Figure 5 viruses-12-00241-f005:**
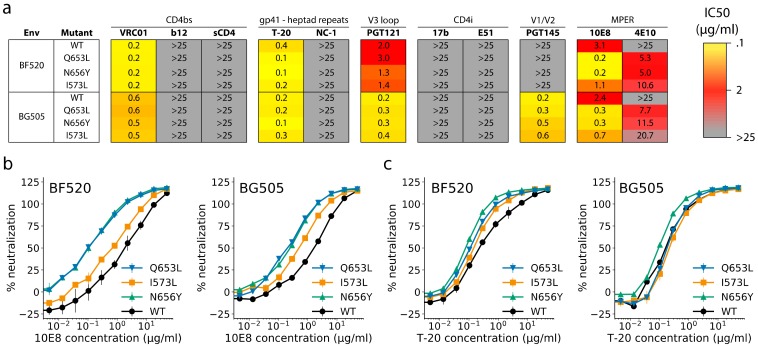
Neutralization of BF520 and BG505 point mutants. (**a**) IC50 values of the indicated drugs and mAbs with BF520 and BG505 Env point mutants. Values report the mean of two independent replicates and are representative of at least two distinct experiments. (**b**,**c**) Neutralization curves illustrating the effects of Env point mutants on neutralization by 10E8 (**b**) and T-20 (**c**). Each data point indicates the mean of two independent replicates and error bars report standard deviation.

**Figure 6 viruses-12-00241-f006:**
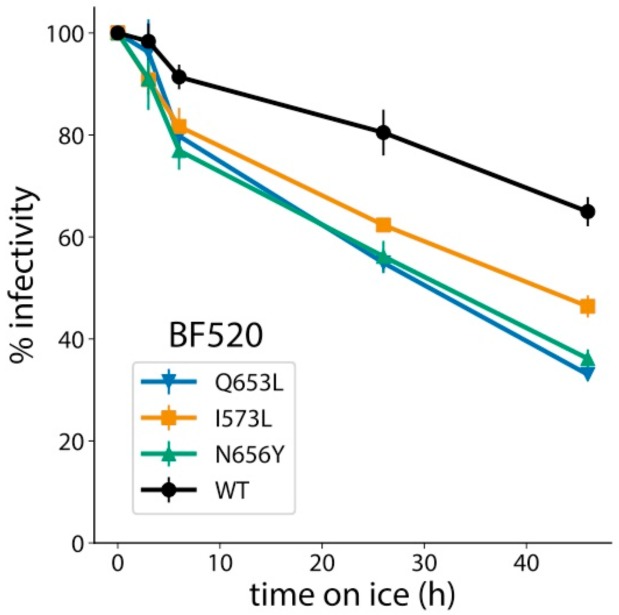
Cold inactivation of BF520 mutant viruses. Viruses were incubated on ice for the indicated period of time then used to infect TZM-bl reporter cells. Percent infectivity of wild-type and mutant strains are shown as a function of time incubated on ice. Each data point indicates the average of four independent replicates and error bars report the standard deviation. Infectivity of viruses not incubated on ice was set to 100%.

**Figure 7 viruses-12-00241-f007:**
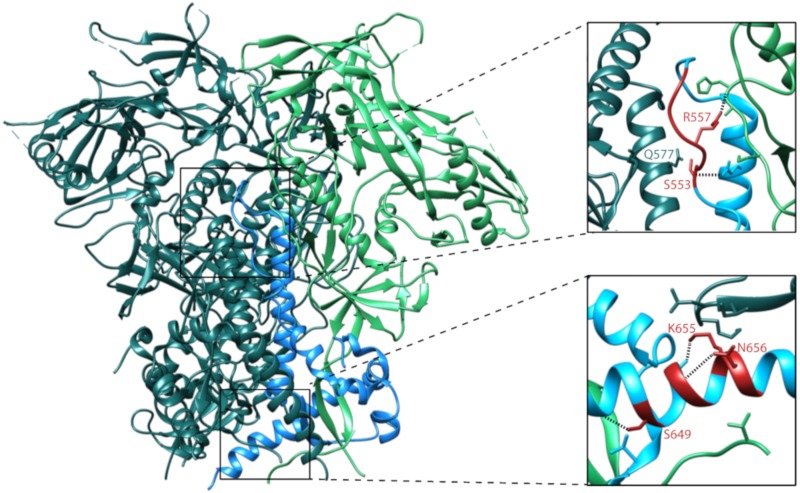
Location and predicted interactions of N-terminal heptad repeat (NHR) and C-terminal heptad repeat (CHR) mutations in the unliganded trimer. The BG505 SOSIP trimer structure (PDB 5CEZ) is shown. Gp120 and gp41 residues of one protomer are shown in bright green and blue respectively. Residues from other protomers are shown in dark green. Upper and lower insets display the regions surrounding the NHR and CHR of one gp41 subunit, respectively. In each inset, residues strongly enriched during the DMS are shown in red. N/CHR residue side chains are shown for amino acids that form predicted H-bond contacts with neighboring residues, and H-bonds are depicted with dotted lines. Side chains are shown for all residues on adjacent subunits that are within 4 angstroms of highly enriched N/CHR residues.

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
