# Peer review of "Identification of HIV-1 Envelope Mutations that Enhance Entry Using Macaque CD4 and CCR5"

_viruses, 2020, doi:10.3390/v12020241_

Round 1

Reviewer 1 Report

Using deep mutational scanning, Roop and colleagues identified mutations in HIV-1 Env that increase entry into HEK293T cells expressing human and/or macaque CD4 and CCR5. Most of these mutations are located in the N- or C-terminal heptad repeat regions of Env. Three of them also increased sensitivity to cold inactivation and neutralizing antibodies targeting the MPER epitope. While the effects of the mutations on HIV-1 entry and Env neutralization are largely convincing, a few controls need to be added. As outlined below, I also suggest to rephrase some of the authors’ conclusions to further improve the quality of the manuscript.

Major points:

Figure 1, lines 94/95: The authors selected “populations expressing intermediate levels of either rhmCCR5 (293TrhmCCR5) or huCCR5 (293ThuCCR5)”. Primary flow cytometry data should be included to demonstrate that expression levels of human and macaque CD4 and CCR5 are indeed similar in the two cell lines used.

Table 1, lines 159-161: Does Table 1 show the results of one exemplary experiment or the mean values of all four replicates? Since the authors do not show any error bars, the reproducibility of the findings remain unclear. Are the effects on T-20 and MPER antibody sensitivity statistically significant?

Fig. 4: Which of the effects are significant? Can the authors perform statistics here? A few exemplary primary flow cytometry data should be included to illustrate the gating strategy.

Fig. 6: How many experiments are shown? What do the error bars indicate?

Lines 446-448: The authors suggest that “the degree to which N/CHR mutants enhance usage of macaque receptors is correlated to increased cold sensitivity and exposure of MPER epitopes on the unliganded Env trimer.” However, N656Y fails to enhance BG505-mediated entry into 293Trhm cells (Fig. 4C), although it increases sensitivity of BG505 to antibodies 10E8 and 4E10 (Table 1). This should be discussed.

Could some of the mutations affect expression, stability or virion incorporation of Env? This should at least be discussed, but may ideally be experimentally addressed.

Lines 460-462: “N/CHR mutations enhanced entry into cells bearing macaque receptors by a second transmitted Env strain, indicating that their effects were not strain specific”. This is not always the case. N656Y enhanced entry in the context of BF520, but had no marked effect in the BG505 backbone. Furthermore, R557L slightly enhanced BF520-mediated entry, but had the opposite effect in the context of BG505 (Figs. 4a, c).

Minor points:

Lines 87/88: CD4 has recently been shown to be highly polymorphic in chimpanzees (PMID: 30718403). How conserved are macaque CD4 and CCR5? It may be helpful to include the accession numbers of the CCR5 and CD4 sequences used in the present study.

Lines 105-111: The origins and subtypes of Q23, BG505.W6.C2.T332N and BF520.W14.C2 initially remain unclear. It may be helpful to provide some background information in this section. Furthermore, the authors should clearly state that these clones represent transmitted/founder viruses.

Lines 200/201: Is the number of 12,559 possible mutations correct if residues 32 to 703 were mutated?

Figures 3 and 5 show residues 32-702, while the respective legends state that residues 32-792 are shown (lines 275, 310). This should be corrected.

Line 262: A table listing the 29 mutations and their effects on 293Trhm and 239Thu infection may be helpful to illustrate species-specific effects.

Lines 287/288: “CHR mutations Q652F and Q653L improved infection of 293Trhm cells 9- and 7-fold respectively”. Are these numbers correct? According to Fig. 4A, Q653L had a more pronounced effect than Q652F.

Line 370: “In these data, we observed a similar drop in frequencies of S375Y, G312V, and A281T”. Can the authors provide numbers here? Was A204E also tested in this study?

Reviewer 2 Report

SHIVs carrying HIV Env from primary isolates do not replicate well in non-human primates primarily due to inefficient entry. The authors have used a deep mutational scanning (DMS)-based screening and identified several mutations in the gp41 regions that facilitate replication of HIV-1 with Env from primary isolates in cells expressing rhesus (rh) CD4 and CCR5.

These findings are novel and interesting. However, there are several points that need to be addressed in order to reveal the relevance of these findings.

Major concerns:

The authors have tested replication efficiency of the Env mutants only in one cell type (293T expressing rhCD4 and rhCCR5). It is possible that the enhancement was specific to this cell type. The authors should test replication of these mutants in other relevant cell types e.g. T cell lines expressing rhesus receptors and in rhesus PBMCs using SIV particles pseudotyped with HIV Env mutants. The authors showed enhanced replication capacities in 293T at high MOI, but only 10% of the cells (at most) were infected. It is unclear if these mutants can replicate in relevant systems.

There are no characterization data for the cell line expressing rhCD4/CCR5 that were generated in this study. Many groups have reported that CD4 and CCR5 surface density is extremely important for the entry of HIV-1 primary isolates and transmitted/founder (T/F) viruses. Are the expression levels of surface rhCD4 and rhCCR5 comparable to those on rhesus PBMCs?

The authors claim the enhancement of the replication of new mutants was due to enhanced entry. Although this is likely the case, there are no data directly supporting the claim. The authors may show ELISA or SPR data that confirm weaker or stronger binding of the mutants to rhCD4. Or a conventional BlaM-based assay to examine entry/fusion efficiency or kinetics may be tested. Something that directly addresses the mechanisms of replication enhancement would be helpful.

More description is needed for the neutralizing assay (Table1). How did you choose the antibodies? How we can read the heat map (unit is missing). Sanders et al. have demonstrated that BG505.T332N is sensitive to the neutralization by CD4-IgG2 in TZMbl [PMID: 24068931]. What makes the differential observation in this study and the one from them? CD4-IgG vs sCD4?

There are no characterization data on pseudotyped virions. The Env content may impact on replication capacity.

Minor points:

Line 19 (abstract); which clone induced 38-fold increase in entry? Is the number from the Fig.4?

Line 325/326; position 599 must be 589.

Discussion is redundant. Some statements are simple repetition of the results, for example the third paragraph (Lines 479-486).

Author Response

Major Points:

Reviewer text in black, our responses in red.

 The authors have tested replication efficiency of the Env mutants only in one cell type (293T expressing rhCD4 and rhCCR5). It is possible that the enhancement was specific to this cell type. The authors should test replication of these mutants in other relevant cell types e.g. T cell lines expressing rhesus receptors and in rhesus PBMCs using SIV particles pseudotyped with HIV Env mutants. The authors showed enhanced replication capacities in 293T at high MOI, but only 10% of the cells (at most) were infected. It is unclear if these mutants can replicate in relevant systems.

We agree that this is a limitation of our study and an interesting direction for additional research. However, as outlined in our cover letter, we feel strongly that these questions are beyond the scope of our work, and would be best addressed in subsequent studies. We now explicitly address these points as limitations in the discussion section. The relevant text is copied below:

A notable limitation of the experiments we performed is that we characterized N/CHR mutants in engineered 293T cells lines that may express CD4 and CCR5 at different levels than human and macaque primary cells. The phenotypes of the Env mutants described here may therefore differ during infection of other cell types, and fitness tradeoffs associated with these mutations may arise during multi-cycle replication in a more clinically relevant context. Additionally, while our data suggest that N/CHR mutations enhance intrinsic Env reactivity, a mechanistic understanding of the effects of these mutations will benefit from further investigations into their effects on CD4 binding and fusion kinetics during infection of relevant cell types.

There are no characterization data for the cell line expressing rhCD4/CCR5 that were generated in this study. Many groups have reported that CD4 and CCR5 surface density is extremely important for the entry of HIV-1 primary isolates and transmitted/founder (T/F) viruses. Are the expression levels of surface rhCD4 and rhCCR5 comparable to those on rhesus PBMCs?

We now provide flow cytometry data in Figure S1 that describe CD4 and CCR5 expression on the engineered cell lines. As with the previous comment, we agree that determining how these expression levels correspond to those of rhesus PBMCs and how this affects selection of Env mutants is an interesting avenue of additional research, but again we feel this is beyond the scope of our current work. We now address this limitation in the discussion section, together with the limitations addressed above.

The authors claim the enhancement of the replication of new mutants was due to enhanced entry. Although this is likely the case, there are no data directly supporting the claim. The authors may show ELISA or SPR data that confirm weaker or stronger binding of the mutants to rhCD4. Or a conventional BlaM-based assay to examine entry/fusion efficiency or kinetics may be tested. Something that directly addresses the mechanisms of replication enhancement would be helpful.

As with the previous requests, we feel that these experiments are best addressed in a follow up work that more explicitly addresses the mechanisms underlying the effects of the mutations identified and validated in this study. This is now stated in the discussion section, together with limitations addressed above.

More description is needed for the neutralizing assay (Table1). How did you choose the antibodies? How we can read the heat map (unit is missing). Sanders et al. have demonstrated that BG505.T332N is sensitive to the neutralization by CD4-IgG2 in TZMbl [PMID: 24068931]. What makes the differential observation in this study and the one from them? CD4-IgG vs sCD4?

The antibodies were chosen so as to interrogate Env regions of specific interest given our mutations and interest in CD4 binding (CD4bs, CD4i, heptad repeats, MPER ) as well as to assess Env regions that would reveal large-scale trimer conformational changes (V3, V1/V2). We now mention this in the results section and the relevant text is copied below:

Drugs and mAbs in the panel were chosen so as to detect alterations to trimer structure in regions both proximal and distal to the mutant N/CHR residues, with particular interest in the CD4 binding site

Previous studies (PMID: 1727487, 2395859) report weaker binding and less potent neutralization of primary HIV isolates by sCD4 as compared to CD4-IgG, which likely explains the discrepancy between our data and that of Sanders et al., 2013.

There are no characterization data on pseudotyped virions. The Env content may impact on replication capacity.

The discussion now mentions the possibility that the mutants we tested impact the Env content of viruses and the potential for this to impact infection. The relevant text is copied below:

Alternatively, N/CHR mutations could enhance entry into macaque cells by a distinct mechanism. For example, mutations to Env’s gp41 and heptad repeat regions have previously been found to alter virus infectivity by altering Env expression, incorporation into virions, and trimer stability. The N/CHR mutations identified in our DMS may enhance entry into 293Trhm though these or other mechanisms, either independent of, or in conjunction with their potential effects on trimer reactivity.”

Minor Points:

Line 19 (abstract); which clone induced 38-fold increase in entry? Is the number from the Fig.4?

This number referred to the fold enrichment following our DMS. We see that this is confusing and so we have re-worded the abstract to reflect the data generating during the pseudovirus infections, shown in Figure 4.

Line 325/326; position 599 must be 589.

This has been fixed.

Discussion is redundant. Some statements are simple repetition of the results, for example the third paragraph (Lines 479-486).

We have removed/consolidated several points in the results and discussion sections so as to be less redundant.

Reviewer 3 Report

Rhesus macaques infected with SIV/HIV chimeras (SHIV) are one of the main animal models in AIDS research and vaccine development. However, SHIVs encoding Envs from circulating HIV-1 strains replicate poorly in macaque cells. Evolved and chronic stage Envs display altered structural conformations and increased susceptibility to neutralizing antibodies compared to Envs from circulating HIV-1 strains. Use of these Envs in SHIVs therefore compromises the relevance of the SHIV/macaque model. The discovery of mutations that allow HIV-1 to use the macaque CD4 receptor without altering neutralization susceptibility has opened the possibility of improving the relevance of the SHIV/macaque model. In this manuscript Roop et al use deep mutational scanning (DMS) to identify novel Env mutations that enhance rhesus macaque CD4 and CCR5 usage. The experiments seem to be well done and the results are consistent. However, I have a few minor concerns:

Figure 2b, 3 and 5: The font size is too small. Figures 3 and 5 are specially difficult to read (I have been unable to distinguish most mutations)

Figure 2a: in the figure it is indicated that after low MOI passage of these pool of  viruses were used to infect 293Thu cells or transfect 293Trhm although the legend indicates that virus pools were then used to infect either 293Trhm or 293Thu cells.

Lines 242-243: Figure 2d mentioned in the text before Figure 2c

Table 1: the units of the concentrations used must be indicated (M, nM, mg/ml, microg/ml, …). Although in the materials and methods section it is indicated that the reported values reflect the mean of four independent replicates, I think this information should be also included in the table legend.

For some experiments there is no indication regarding assay reproducibility and the number of times they were repeated (Figure 1 and 6). This information should be included in the figure legends indicating whether the results are the mean of X experiments or one representative experiment of X.

Line 493: protomors (protomers?)

Why is the structure of the BG505 SOSIP.664 trimer in complex with a PGT121 precursor Fab used for modeling Env mutants? Although this Ab bound structure forms a near-native trimer structure, bound antibodies could influence conformation. Why not using the BG505 SOSIP ligand free structure? Also, although I think this does not affect the conclusions reached in this manuscript, care should be taken when using BG505 SOSIP.664 structure for modeling mutations as there is some controversy of whether the changes introduced to stabilize the trimer might alter Env conformation stabilizing state 2 or a state 2-like conformation instead of state 1.

Some references are not correctly formatted:  for example reference 11 (authors) and 12 (journal). Also, some references are not numbered in the order of appearance in the text (the order in the text is 1-19, 28, 50-54, 21, 20, 22-49)

Author Response

Figure 2b, 3 and 5: The font size is too small. Figures 3 and 5 are specially difficult to read (I have been unable to distinguish most mutations)

We have darkened the color of the font in future 2b (now Figure 1b) to make it easier to read, and resized the logo plot figures to make the mutations easier to see.

Figure 2a: in the figure it is indicated that after low MOI passage of these pool of  viruses were used to infect 293Thu cells or transfect 293Trhm although the legend indicates that virus pools were then used to infect either 293Trhm or 293Thu cells.

“Infect” is the correct word in all cases. This has been fixed.

Table 1: the units of the concentrations used must be indicated (M, nM, mg/ml, microg/ml, …). Although in the materials and methods section it is indicated that the reported values reflect the mean of four independent replicates, I think this information should be also included in the table legend.

We now include this information.

For some experiments there is no indication regarding assay reproducibility and the number of times they were repeated (Figure 1 and 6). This information should be included in the figure legends indicating whether the results are the mean of X experiments or one representative experiment of X.

We have included this information in all figure captions in which it was not previously given.

Line 493: protomors (protomers?)

This has been fixed.

Why is the structure of the BG505 SOSIP.664 trimer in complex with a PGT121 precursor Fab used for modeling Env mutants? Although this Ab bound structure forms a near-native trimer structure, bound antibodies could influence conformation. Why not using the BG505 SOSIP ligand free structure? Also, although I think this does not affect the conclusions reached in this manuscript, care should be taken when using BG505 SOSIP.664 structure for modeling mutations as there is some controversy of whether the changes introduced to stabilize the trimer might alter Env conformation stabilizing state 2 or a state 2-like conformation instead of state 1.

We assume the reviewer is referring to the unliganded BG505 structure obtained by Kwon et al., (PDB 4ZMJ). We considered using this structure but decided against it, as the majority of NHR residues are not resolved, including those where we see the largest effect mutations in our DMS data.

We agree with the reviewer that great care should be taken in drawing conclusions from the PGT121 bound trimer, especially given the stabilizing mutations introduced into SOSIP trimers. For this reason and others we have made sure to indicate that our structural observations as speculative, and have confined them to the discussion section.

Some references are not correctly formatted:  for example reference 11 (authors) and 12 (journal). Also, some references are not numbered in the order of appearance in the text (the order in the text is 1-19, 28, 50-54, 21, 20, 22-49)

References are now correctly formatted.

Round 2

Reviewer 1 Report

In the revised version of their manuscript, the authors addressed all of my remaining concerns. Among other things, they included primary flow cytometry data (new Fig. S2) demonstrating that expression of CCR5 and CD4 receptors was similar (but not identical) between 293Trhm and 293Thu cell types. Furthermore, they now show neutralization curves in new Fig. 5. Finally, they have expanded their description of the HIV-1 strains used in this study and cautioned some of their conclusions.

I just have two very minor (optional) points that the authors may want to address before publication:

  • While an accession number for macaque CCR5 has been provided in the revised manuscript, I suggest to also include accession numbers for human CCR5, macaque CD4 and human CD4 and/or deposit the sequences they used.
  • While the authors explain the gating strategy in their response letter, I still feel that it would be nice to include a few exemplary primary FACS data in this figure. This would help to assess the quality of the data particularly since the infection rates are <1% in some of the cases.

Reviewer 2 Report

The revised manuscript has largely addressed the comments from reviewers. I have no further concerns.